# Classification of Alzheimer’s Progression Using fMRI Data

**DOI:** 10.3390/s23146330

**Published:** 2023-07-12

**Authors:** Ju-Hyeon Noh, Jun-Hyeok Kim, Hee-Deok Yang

**Affiliations:** Department of Computer Engineering, University of Chosun, Gwangju 61452, Republic of Korea; narak@chosun.kr (J.-H.N.); joonhyeok10@chosun.kr (J.-H.K.)

**Keywords:** Alzheimer’s disease, 3D U-Net, deep learning

## Abstract

In the last three decades, the development of functional magnetic resonance imaging (fMRI) has significantly contributed to the understanding of the brain, functional brain mapping, and resting-state brain networks. Given the recent successes of deep learning in various fields, we propose a 3D-CNN-LSTM classification model to diagnose health conditions with the following classes: condition normal (CN), early mild cognitive impairment (EMCI), late mild cognitive impairment (LMCI), and Alzheimer’s disease (AD). The proposed method employs spatial and temporal feature extractors, wherein the former utilizes a U-Net architecture to extract spatial features, and the latter utilizes long short-term memory (LSTM) to extract temporal features. Prior to feature extraction, we performed four-step pre-processing to remove noise from the fMRI data. In the comparative experiments, we trained each of the three models by adjusting the time dimension. The network exhibited an average accuracy of 96.4% when using five-fold cross-validation. These results show that the proposed method has high potential for identifying the progression of Alzheimer’s by analyzing 4D fMRI data.

## 1. Introduction

As the most common form of age-related dementia [1,2], Alzheimer’s disease (AD) is a neurodegenerative disorder characterized by memory loss, impaired thinking, and behavioral problems. It manifests due to disturbances in communication between different regions of the brain [3]. AD often leads to the death of nerve cells and loss of brain tissue, with the brain shrinking and malfunctioning over time [4]. Immediately prior to AD, the brain is subject to mild cognitive impairment (MCI) [5]. It has been determined that 15% of the population over 65 years of age already has MCI, and more than half of those individuals suffer from AD after five years [6]. The brain states leading up to AD can be divided into three stages: EMCI, LMCI, and AD. To realize preventative measures, AD must be predicted at early stages. Accordingly, several prior studies have focused on predicting the initial state.

In the field of healthcare, advances in magnetic resonance imaging (MRI) have enabled the efficient observation of brain conditions, making more information available for early-stage AD prediction. Structural MRI has revealed atrophy in the medial temporal and cingulate cortices, as well as more widespread pathology. Eskildsen et al. [7] used the thickness of the brain cortex to identify the transition between MCI and AD with 80% accuracy. Beheshti et al. [8] distinguished AD patients from healthy controls (HCs) using voxel-based morphometric techniques to extract global and local gray matter. AliHaidr Syaifullah et al. [9] obtained a result of 90.5% in the diagnosis of AD using support vector machines (SVM) on a limited dataset. Lolis nanni et al. [10] successfully identified AD and CN cases with an accuracy of 90.8% using transfer learning. They used a 3D convolutional neural network (CNN) model and conducted experiments using AlexNet, GoogleNet, and ResNet among other datasets. Furthermore, they conducted experiments on MCI convert AD (MCIc) vs. MCI not-convert AD (MCInc) and MCIc vs. CN, achieving accuracies of 71.2% and 84.2%, respectively. Nikhil J. Dhinagar et al. [11] utilized the vision transformer (ViT), a prominent model in the field of computer vision, to detect AD from sMRI data. They achieved an accuracy of 89%, demonstrating the capability of ViT to process brain images.

Several studies have used rest-state functional MRI (rs-fMRI), which has demonstrated extremely high sensitivity for AD [12]. Consequently, rs-fMRI has been reported to exhibit high performance in distinguishing AD [7]. Using rs-fMRI, Grieder et al. [13] found that a decline in neural connection complexity is directly related to AD. This result verifies the relationship between MCI and AD, as connection complexity begins to decline with the former. Additionally, rs-fMRI has been reported to show functional connectivity associated with cognitive impairments in aging populations with health problems, MCI, and AD. Each stage must be distinguished according to changes in neural complexity.

In the field of neurology, various deep learning models have been deployed to analyze fMRI data. Typically, analyses to distinguish between AD and CN states are conducted using a CNN model [7,14,15,16,17]. However, 2D fMRI data has generally been used for binary classification. Further research must be conducted on the multi-class classification of 2D fMRI data.

Unlike previous studies, we propose a deep learning network that considers spatiotemporal features to identify the decreasing interregional connectivity complexity of the brain and simultaneously distinguish brain states. The temporal features are obtained from rs-fMRI, which best identifies changes in brain networks. We employed a 3D-CNN architecture based on U-Net to extract spatial characteristics. By conserving the time axis, this architecture allows for an output with the same shape as the input, which is ideal for extracting temporal information. Because recurrent neural networks (RNNs) represent the best approach for processing time-series data, we employed an RNN structure to extract temporal features using this conserved time information. The long short-term memory (LSTM) algorithm is used to process one of several RNN structures, and the fully connected layers are then used for classification. Thus, we constructed a model that achieves an accuracy of 96.4%. Furthermore, we conducted experiments by utilizing 1 × 1 convolution to adjust the temporal dimension, halving it and making adjustments at quarter intervals.

## 2. Related Works

### 2.1. Alzheimer’s Disease

AD is an irreversible neurodegenerative disorder that has recently emerged as part of the human lifespan. At the time of its initial identification in 1906 by the German doctor Alois Alzheimer, the disease had a relatively low prevalence with only a small number of individuals affected by it. Today, it is the leading cause of dementia in 10% of individuals over 65 and half of those over 85. Currently, there are no medical treatments that can fully cure or prevent AD; its progression can only be slowed. Accordingly, it is critical to detect AD early and begin treatment to slow its progression. It is imperative to detect MCI, also known as the pre-dementia stage, early and initiate appropriate treatment. Although MCI is known to cause mnemonic and cognitive decline, 10–15% of patients progress to AD every year with no significant disruption to their daily lives. This indicates that the early detection of MCI can slow its progression to AD [12].

Among the various approaches to diagnosing AD, fMRI is a non-invasive methodology that can be used to diagnose the brain with the highest detail. rs-fMRI is an fMRI method that measures functional brain activity and neurological changes by photographing the brain over a set period [15]. This technique enables the identification of brain regions with functional connectivity to specific regions in the dormant brain (seed-based functional connectivity analysis and independent component analysis), understanding of the characteristics of multiple brain regions (graph-based network analysis), and interpretation of spontaneous activity patterns (local homogeneity, low-frequency vibration analysis, and Hurst index analysis). Many attempts have been made to identify neurological states with a focus on the ability to analyze brain connectivity.

### 2.2. U-Net

The U-Net [18] model, which won the 2015 ISBI Cell Tracking Challenge by a wide margin, has demonstrated accurate image segmentation performance with a minimal volume of learning data. The encoder–decoder structure obtains features at each layer, increasing the number of channels and decreasing the dimensionality during the encoding process. This is referred to as the contracting path. Only low-dimensional encoded information is used in the decoding step to decrease the number of channels and increase dimensionality, thereby restoring a high-dimensional image. This is known as the extended path. The configurations of the reduction and extension paths are symmetrical. The encoder–decoder-structured network loses precise position information on the image object during the encoding stage owing to dimensionality reduction, and it is unable to compensate for this loss during the decoding stage because it only employs low-dimensional information. The fundamental idea behind U-Net is to position images accurately while extracting their attributes by utilizing both low- and high-dimensional information. To achieve this, we concatenated the features from each layer of the encoding stage into each layer of the decoding stage, with the layers linked via skip connection. Figure 1 represents the architecture of U-net.

### 2.3. Time-Series Network

The RNN—a deep learning network that produces results utilizing continuous data, or data with a time axis—performs at the greatest level. This architecture is an example of an artificial neural network wherein circulating structures are created by connecting hidden nodes to edges in certain directions.

The input from the time step may be received as shown in Figure 2, and the outcome may be modified for use. However, if backpropagation is performed throughout each time step, a large time step significantly increases network depth, which poses the issues of vanishing and expanding gradients. Another problem is that long-term patterns cannot be learned.

Although the RNN and LSTM both adopt a chain structure, the LSTM’s module has a structure in which four layers exchange information. To determine what to remember, input, forget, and output gates are added to the memory cell by altering the hidden layer of the RNN. Compared with conventional RNNs, LSTM demonstrates outstanding performance when processing lengthy input sequences.

fMRI data are time-series data. Owing to the above characteristics, Gao et al. employed an RNN to analyze fMRI data and estimate age [19]. Li et al. used LSTM to identify brain states associated with specific events by analyzing connectivity [20]. Parmer et al., who inspired our study, used LSTM alongside a CNN to classify CN, EMCI, LMCI, and AD [21]. The present study utilized LSTM because of its proven high performance in interpreting the temporal characteristics of fMRI data.

## 3. Materials and Methods

### 3.1. Pre-Processing

We used the online Alzheimer’s Disease Neuroimaging Initiative (ADNI) online database [22]. In ADNI, resting-state fMRI volumes were acquired for 699 subjects distributed among four classes—CN, EMCI, LMCI, and AD—in the ADNI2 phase. Table 1 lists the number of participants in each class.CN—108F/89M, age: 65–96EMCI—142F/96M, age: 56–90LMCI—58F/101M, age: 57–88AD—56F/62M, age: 56–89

The flip angle was 80° with a repetition time of 3000 ms and an echo time of 1 ms. A total of 140 functional volumes were acquired. Each volume had 48 axial slices with a thickness of 3.313 mm. Two normalization methods were applied: direct and indirect. Indirect normalization requires sMRI images, whereas direct normalization—which we adopted in this study—uses only fMRI images. The pre-processing constitutes four steps, and it was performed using the MATLAB SPM12 toolbox. In the first step, as shown in Figure 3a, samples were realigned to eliminate noise caused by head movement. Specifically, rigid body registration was performed to correct the signal values by considering the translation or rotation of the head position during MRI. The next step, slice timing, corrected any discrepancies in the time of sample acquisition. Subsequently, normalization was performed to convert the brain images of all participants into standardized shapes to facilitate individual comparisons. We set the voxel size parameter to (2, 2, 2). Finally, smoothing was performed to correct the value of each voxel by averaging the value of a neighboring voxel. Thus, the signal-to-noise ratio (SNR) was improved by eliminating the high-frequency band. Generally, these values are corrected using a Gaussian full width at half maximum (FWHM) kernel. These pre-processing steps were performed using the functions provided by SPM12. Subsequently, fMRI was performed to obtain an image of (79, 95, 79, 140). Because this shape is not appropriate for our model’s input, it was converted to (128, 128, 128, 140).

### 3.2. Model

fMRI images represent 4D data in 3D space with a 1D time axis. Considering these characteristics, the proposed method consists of two steps. The 3D-CNN first extracts spatial characteristics and then passes them to the LSTM model to extract temporal characteristics. We employed the U-Net architecture to extract spatial features. The execution was performed by changing the time axis of the data between three sizes in consideration of the dataset’s size. The lengths of the time axes were initially 140 and changed to 70 and 35 using a 1 × 1 convolution. Following the extraction of spatial features, LSTM was used to process 140, 70, and 35 functional volume time-series data points containing temporal information. Temporal features were extracted using LSTM, and classification was performed using a fully connected layer. The final layer employed the softmax function as the active function.

### 3.3. Spatial Feature Extraction

The 3D-CNN used to extract spatial features was constructed using the U-Net model based on an encoder–decoder structure [12]. A normal encoder–decoder-based network reduces the dimensions in the encoding step and restores them in the decoding step to generate high-dimensionality output. However, when dimensionality is reduced by the encoder, detailed location information about the image is inevitably lost. The encoding and decoding layers are connected using skip connections, and each layer result is concatenated, as in U-Net. Figure 4 shows spatial feature extractor architecture.

Our model input is 4D data containing a time axis. We considered three approaches to handle inputs: keeping the time axis intact and using 1 × 1 convolutions to reduce the time axis by 2 and 4, as in Parmar et al. [17]. Table 2 lists the model layers used in this study. During downsampling, the image size was halved by setting the stride to 2 for convolutional operations. Conversely, during upsampling, the image size was restored using transpose convolutional operations with a stride set to 2. As shown in Table 2, each model’s symmetrical structure was adjusted to have 140, 70, or 35 channels in the last layer to maintain temporal information. The depth of each model was slightly different owing to the limitations of computing power. A detailed layer configuration of the spatial feature extractor can be found in Table 3.

### 3.4. Temporal Feature Extractor and Classifier

The rs-fMRI data are 4D data with a time axis obtained by photographing a 3D image during a unit of time. Spatial and temporal features must be considered simultaneously to accurately measure brain conditions. Temporal information is extracted using LSTM, which specializes in time-series data processing. Unlike spatial feature extractors, temporal feature extractors feature a straightforward structure. Each LSTM is adjusted with 140, 70, or 35 channels, and the flatten layer is used for classification. Figure 5 presents the temporal extractor’s structure.

Following the extraction of temporal features, classification is performed using fully connected layers. The first fully connected layer yields an output of 256 shapes resulting from the temporal feature. The second fully connected layer classifies the output between four classes using the softmax function.

### 3.5. Hyperparameters

Five-fold cross-validation was performed during the trial. The dataset was separated into five files, four of which were utilized for training, and one of which was used for validation. Training was conducted evenly in each class. We employed the Adam optimizer [23] as represented by Equations (1)–(3).
(1)$$m_{t}=\beta_{1}m_{t-1}+\left(1-\beta_{1}\right)\nabla_{\omega }J\left(\omega_{t}\right),$$
(2)$$v_{t}=\beta_{2}m_{t-1}+\left(1-\beta_{2}\right){\left(\nabla_{\omega }J\left(\omega_{t}\right)\right)}^{2},$$
(3)$$\omega_{t+1}=\omega_{t}-m_{t}\frac{\eta }{\sqrt{v_{t}+\varepsilon }},$$
where $\varepsilon =1^{-8}$, $\beta_{1}=0.9,$ and $\beta_{2}=0.999$.

Training was performed every 100 epochs with an initial learning rate of 0.00001 that decreased using exponential decay. Dropout was set to 0.2 to avoid overfitting. Based on the GPU NVIDIA RTX A6000, a network configured in this manner required 25 h to learn.

With the exception of pre-processing, all training and testing procedures were performed in a Python environment using TensorFlow. Training required approximately four days to complete over 100 epochs.

## 4. Experimental Results

The experiment was performed with three models with different time axes. The three models are compared and evaluated quantitatively in Section 4.1. In the experiment, five-fold cross-validation was performed to evaluate the performance of each model. Section 4.2 presents and explains the performance metrics. 

### 4.1. Proposed Model

Table 4 displays the accuracy results obtained by applying five-fold cross-validation for each fold. The average accuracy of models with a quarter-reduced time axis was 91.8%. In contrast, models with a half-reduced time axis exhibited an average of 95.22%, which suggests that many properties included on the time axis were lost with a significant reduction. Models without a time-axis reduction achieved an average accuracy of 96.28%. Although it required a relatively lengthy learning process, the 140-channel model exhibited the highest accuracy for each class. All models obtained good representations of AD and CN; however, when there were fewer channels, EMCI and LMCI performed poorly, indicating that a more advanced network is required to distinguish between the two. Such a network must have a time-resolution effect to detect slight variations between two classes.

### 4.2. Performance Metrics

Figure 6 represents the results using a confusion matrix according to the time associated with each model’s peak performance. Performance was evaluated using precision, recall, F-score, and accuracy, computed using Equations (4)–(7). Although the 140-channel model performed best across the board, it required a slightly longer learning time than the other models. In comparison, the accuracy of the 70-channel model decreased by 0.72%, its precision and recall decreased by 0.48% and 1.98%, and its F1-score decreased by 1.37%. Differences in accuracy, precision, recall, and F1-score across 35 channels were 5%, 5.68%, 5.04%, and 5.37%, respectively. As the time axis was reduced, performance decreased for each evaluation metric.
(4)$$\mathrm{Precision}=\frac{TP}{TP+FP}\;$$
(5)$$\mathrm{Recall}=\frac{TP}{TP+FN}$$
(6)$$F-\mathrm{score}=2\times \frac{Precision\times Recall}{Precision+Recall}$$
(7)$$\mathrm{Accuracy}=\frac{TN+TP}{Total\;number\;of\;samples\left(N\right)}$$

Figure 7 shows the variations in the accuracy and loss in the three models following training. The training process takes longer and becomes more accurate as the number of channels increases. The model with the highest accuracy was saved and used for testing.

In the 35-channel model, the loss graph can be observed to decrease gradually. In comparison, the loss graphs of both other models exhibit a significantly higher decay. Although the 140-channel model exhibits the highest accuracy, the 70-channel model decreases and converges the most quickly. This demonstrates how the resolution of the time axis increases accuracy.

Figure 8 illustrates the last layer of the spatial extractor obtained by compressing the time axis. The input characteristics indicate that the characteristic map is accurate owing to the U-Net structure.

## 5. Discussion

Using fMRI data, we built a network to classify patients into AD, CN, EMCI, and LMCI groups. Although fMRI data are typically normalized to sMRI data, we employed direct normalization. An experiment was conducted with three models using different time-axis settings. Specifically, the model input was reduced by a factor of two or four, and it was restored using a 1 × 1 × 1 kernel. In the spatial feature extractor, the input layer was adjusted according to the time axis, and the result was used as input in the time feature extractor. U-Net was used as the spatial feature extractor, and LSTM was used as the temporal feature extractor. Classification was performed using two fully connected layers with the extracted features. The three models achieved accuracy scores of 96.43%, 95.71%, and 91.43%.

Figure 9 compares our model with a conventional 3D-CNN model. Although the 3D-CNN model initially achieves the highest accuracy, our model ultimately outperforms it. The 3D-CNN model reduced the time dimension to 1 through 1 × 1 convolution, and the experiment was conducted by increasing the channel up to 256 with five convolutional and max-pooling layers.

Figure 10 compares the three models in terms of performance. The 140-channel input model exhibited the highest accuracy at 96.43%. The 70-channel input model obtained an accuracy of 95.71%. The 35-channel input model exhibited a significant decrease in accuracy, with a score of 91.43%. Although a single reduction of the time axis does not significantly impact performance, it can be inferred that accuracy decreases rapidly with subsequent reductions.

Table 5 compares the accuracy of the proposed model with that of existing methods. Although binary classification models obtained accuracy of up to 98.3%, they perform poorly on multi-class problems. The multi-class model with the highest accuracy of 97.6% considers only 2D data. For 4D data, the highest obtained accuracy was 89.4%. Our methodology addresses the multi-class classification problem using 4D data, which exhibits the highest performance for this task. Given the importance of accurately predicting the level of MCI in preventing Alzheimer’s disease, the proposed methodology is the superior prevention tool.

## 6. Conclusions

We performed an experiment using a model for classifying 4D fMRI data related to AD among four classes. Spatial information was extracted using U-Net to utilize 4D fMRI data with temporal-spatial characteristics, and temporal information was extracted using LSTM. Three models with different time-dimension inputs (140 channel, 70 channel, and 35 channel) were used, and their respective classification accuracy scores were 96.43%, 95.71%, and 91.43%. Because our dataset was somewhat small, we believe that better results can be obtained by collecting more data and modifying the network in future experiments.

## Figures and Tables

**Figure 1 sensors-23-06330-f001:**
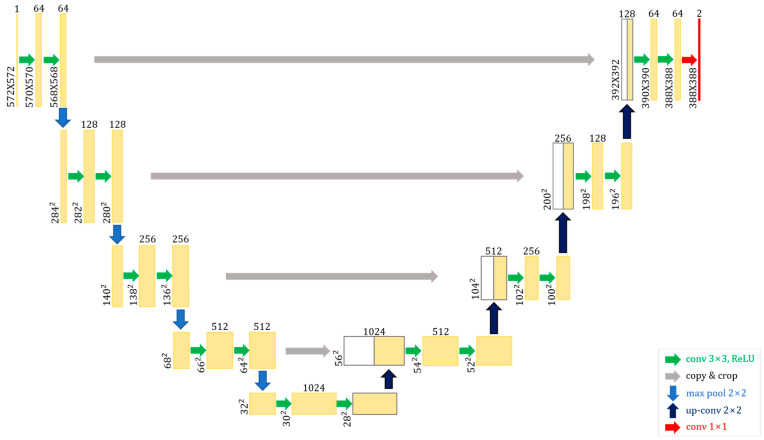
U-Net architecture.

**Figure 2 sensors-23-06330-f002:**
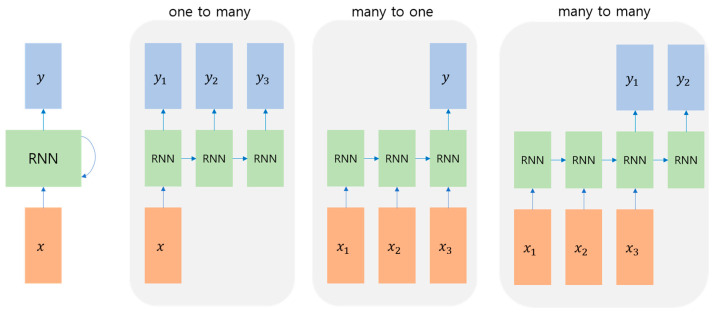
Example of RNN.

**Figure 3 sensors-23-06330-f003:**
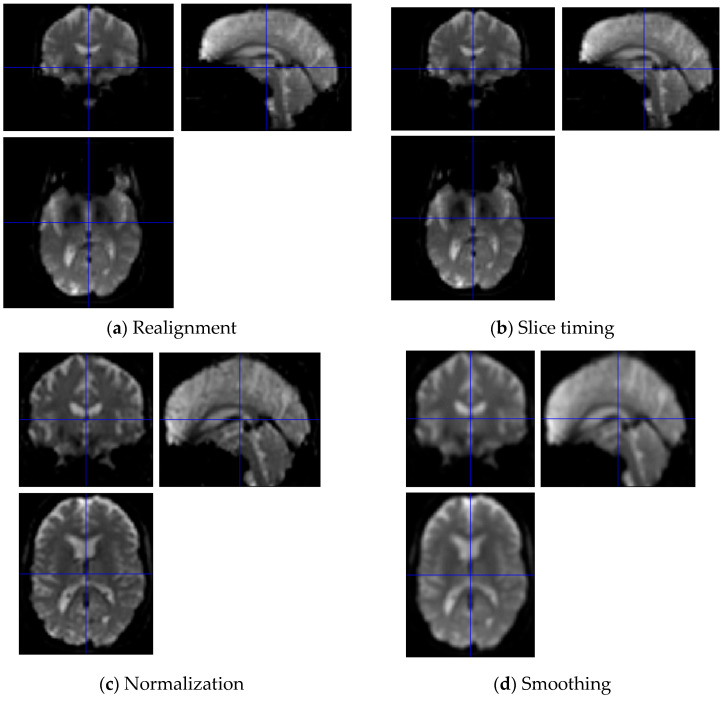
Pre-processing of fMRI data.

**Figure 4 sensors-23-06330-f004:**
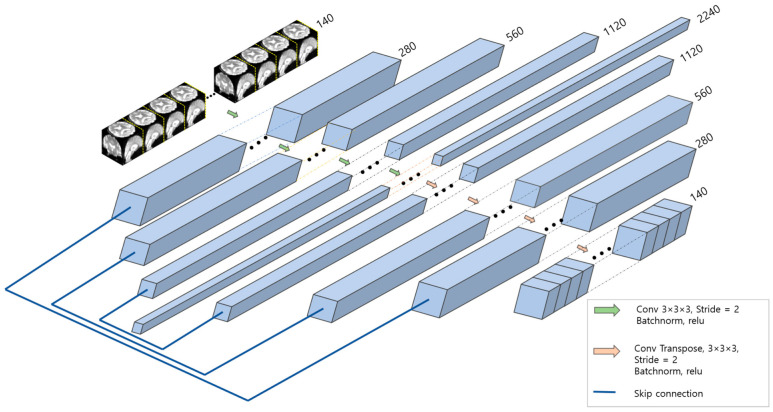
Spatial feature extractor. Downsampling by 3 × 3 × 3 convolution, stride = 2. Upsampling by 3 × 3 × 3 transposed convolution, stride = 2.

**Figure 5 sensors-23-06330-f005:**
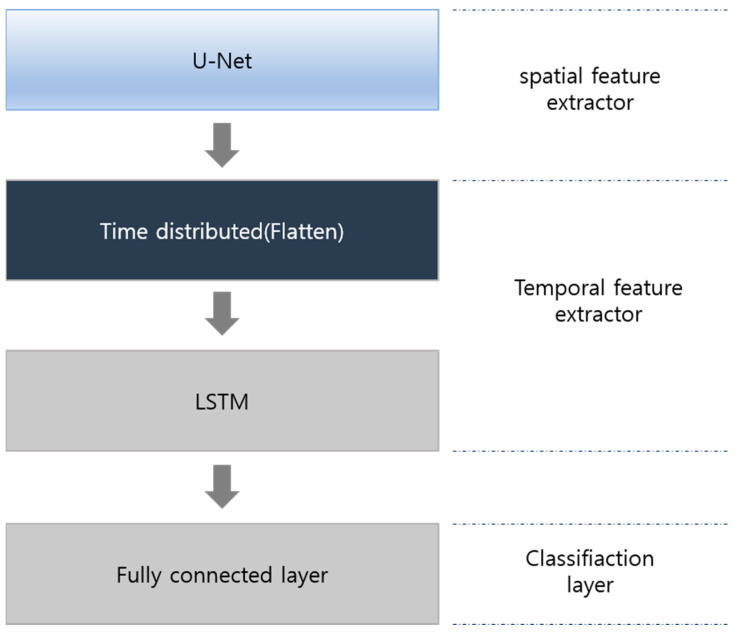
Temporal feature extractor and classifier.

**Figure 6 sensors-23-06330-f006:**
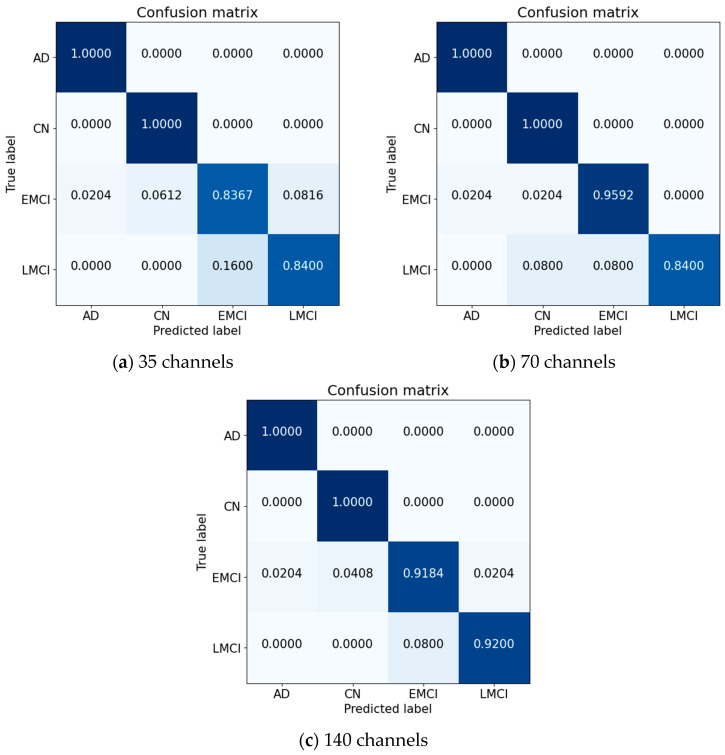
Confusion matrices for three models.

**Figure 7 sensors-23-06330-f007:**
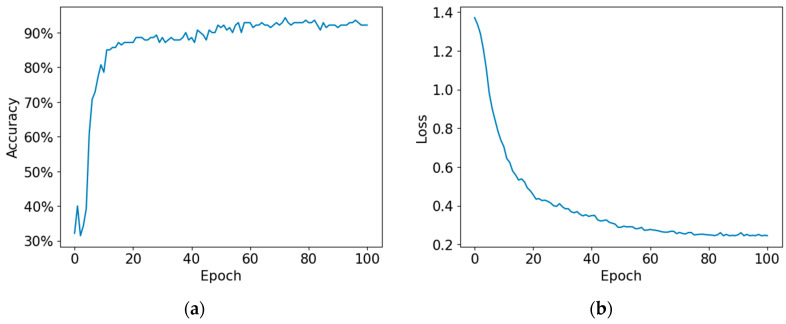
(**a**) 35-channel model accuracy and (**b**) loss. (**c**) 70-channel model accuracy and (**d**) loss. (**e**) 140-channel model accuracy and (**f**) loss.

**Figure 8 sensors-23-06330-f008:**
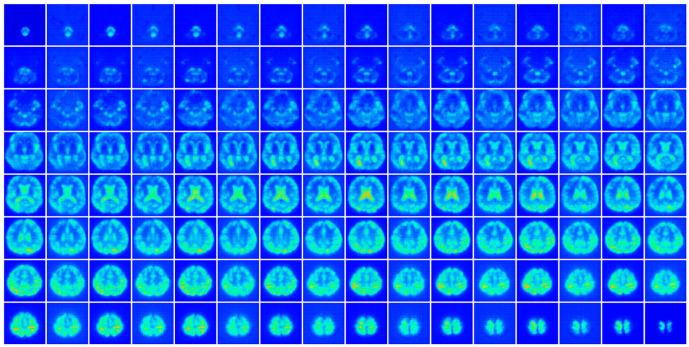
Activation map of the spatial extractor’s last layer. The time axis was reduced using the time-axis mean. The brighter side of an image is an activated feature from the input image.

**Figure 9 sensors-23-06330-f009:**
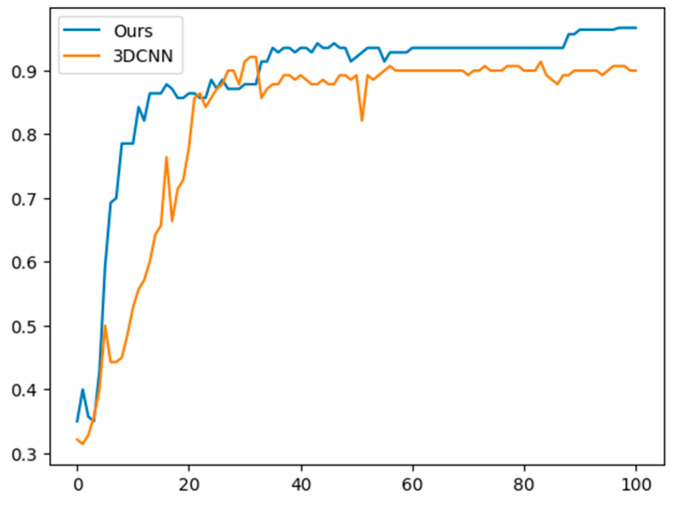
Comparison between proposed model and conventional 3D-CNN model.

**Figure 10 sensors-23-06330-f010:**
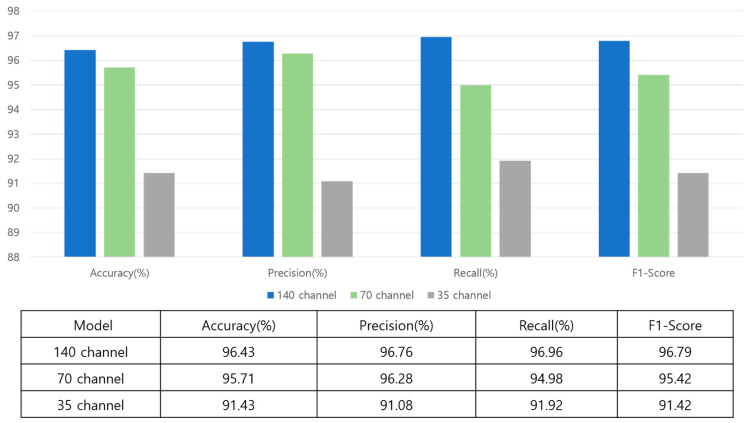
Bar graph showing the performance of three different models.

**Table 1 sensors-23-06330-t001:** Number of subjects in four classes.

CN	EMCI	LMCI	AD
197	238	159	118

**Table 2 sensors-23-06330-t002:** Time-axis layer and output of intact and reduced models. During 1 × 1 convolution, the stride was set to 2, which halved the size of the image. Conversely, when upsampling the image, a transpose convolution with a stride of 2 was used to double the size of the image.

Spatial Feature Extractor	Intact	Reduced 1/2	Reduced 1/4
Input	128 × 128 × 128 × 140	128 × 128 × 128 × 70	128 × 128 × 128 × 35
Layer 1	64 × 64 × 64 × 280	64 × 64 × 64 × 140	64 × 64 × 64 × 70
Layer 2	32 × 32 × 32 × 560	32 × 32 × 32 × 280	32 × 32 × 32 × 140
Layer 3	16 × 16 × 16 × 1120	16 × 16 × 16 × 560	16 × 16 × 16 × 280
Layer 4	8 × 8 × 8 × 2240	8 × 8 × 8 × 1120	8 × 8 × 8 × 560
Layer 5	16 × 16 × 16 × 1120	4 × 4 × 4 × 2240	4 × 4 × 4 × 1120
Layer 6	32 × 32 × 32 × 560	8 × 8 × 8 × 1120	2 × 2 × 2 × 2240
Layer 7	64 × 64 × 64 × 280	16 × 16 × 16 × 560	4 × 4 × 4 × 1120
Layer 8	128 × 128 × 128 × 140	32 × 32 × 32 × 280	8 × 8 × 8 × 560
Layer 9	-	64 × 64 × 64 × 140	16 × 16 × 16 × 280
Layer 10	-	128 × 128 × 128 × 70	32 × 32 × 32 × 140
Layer 11	-	-	64 × 64 × 64 × 70
Layer 12	-	-	128 × 128 × 128 × 35

**Table 3 sensors-23-06330-t003:** Layer Detail. CB unit is convolution and batch normalization. CTB unit is transpose convolution and batch normalization. All convolutions and transpose convolutions proceeded with a stride of 2.

Intact	Reduced 1/2	Reduced 1/4
CB Unit		
CB Unit		
CB Unit		
CB Unit		
CTB Unit	CB Unit	
CTB Unit		CB Unit
CTB Unit		
CTB Unit		
-	CTB Unit	
-	CTB Unit	
-	-	CTB Unit
-	-	CTB Unit

**Table 4 sensors-23-06330-t004:** Five-fold cross-validation accuracy performance of three models.

	140 Channel (%)	70 Channel (%)	35 Channel (%)
Test 1	96.1	95.7	92.4
Test 2	96.4	94.8	92.2
Test 3	95.8	95.4	92.3
Test 4	96.7	95.1	91.6
Test 5	96.4	95.1	91.4
Average	96.28	95.22	91.8

**Table 5 sensors-23-06330-t005:** Comparison of AD classifiers based on fMRI modality between binary and multi-class classification. The binary approach classifies between AD and CN, whereas the multi-class approach classifies between AD, EMCI, LMCI, and CN.

Research	Modality	Type	Accuracy (%)
Sarraf et al. [24]	fMRI-2D	Binary	96.8
Billones et al. [25]	fMRI-2D	Binary	98.3
Jain et al. [26]	MRI-2D	Binary	99.1
Li et al. [27]	fMRI-4D	Binary	97.3
Parmar et al. [21]	fMRI-4D	Binary	94.5
Billones et al. [25]	fMRI-2D	Multi-class	91.8
Kazemi et al. [28]	fMRI-2D	Multi-class	97.6
Li et al. [27]	fMRI-4D	Multi-class	89.4
Harshit et al. [21]	fMRI-4D	Multi-class	94.5
Ours	fMRI-4D	Multi-class	96.4

## Data Availability

In this research, a public dataset was used, which can be found at: http://adni.loni.usc.edu, accessed on 18 May 2022.

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
