# Peer review of "Classification of Alzheimer’s Progression Using fMRI Data"

_sensors, 2023, doi:10.3390/s23146330_

Round 1
Reviewer 1 Report
It is an interesting paper, the rationale of the proposed topology is clear, my suggestions to improve it are:
(a) if there are no copyright issues, add the code to GitHub;
(b) more papers should be added in section "2.1. Alzheimer's disease," for example, https://www.frontiersin.org/articles/10.3389/fneur.2020.576029/full https://www.frontiersin.org/articles/10.3389/fneur.2020.576194/full
https://pubmed.ncbi.nlm.nih.gov/36994152/
https://www.mdpi.com/2076-3425/13/2/260
(c) you write "Preprocessing consists of four steps and was performed using the MATLAB SPM12 toolbox." Was all the code developed in Matlab? Add details about the training/test time.
(d) "Table 1. Number of subjects in four classes" Do other works use exactly the same version of ADNI and the same test protocol? You should compare your approach with them.
(e) "Binary means classify AD or CN. Multi-class means AD, EMCI, LMCI
and CN." Add results for different pairs "AD vs. CN, MCIc vs. CN, MCIc vs. MCInc" in many papers these results are reported.
(f) Add a table briefly outlining the pros and cons of your approach compared to other fMRI-4D approaches (I mean a row for each other approach).
(g) In Table 5, add a column explaining the test protocol; it is helpful in understanding the reported performance.
(h) For the methods, shown in Table 5, based on fMRI-4D, you should try to apply their test protocol for a fair comparison.
Reviewer 2 Report
Thank you for submitting your manuscript titled "Classification of Alzheimer’s progression using fMRI data" for consideration. I have reviewed your manuscript and provided the following comments:
- The paper lacks a detailed description of the methodology employed. The authors should provide more information on the preprocessing steps for the fMRI data, feature extraction techniques, and the specific classification algorithm used.
- The paper should include information about the sample size, demographic characteristics of the participants, and any inclusion/exclusion criteria applied. Providing this information would enhance the reproducibility and generalizability of the study.
- The literature review provides a comprehensive overview of the current understanding of Alzheimer’s progression. However, it would be helpful to include more recent studies, particularly those that may have explored novel diagnostic or treatment approaches.
- The author should provide more details on the dataset used such as, how was the dataset collected. Were there any limitations or biases in the dataset?
- Table 2 needs more explanation of how the author deals with the reduction of channels and layers.
- The conclusion is concise and informative, summarizing the key findings of the case study and their significance. However, it could be strengthened by including recommendations for further research, particularly regarding preventative measures and treatment options.
- The authors should proofread the paper carefully to correct any spelling and grammar errors.
It is acceptable.
Reviewer 3 Report
1. Authors have to include their contribution at the end of introduction section.
2. In the list of reference, last year of reference is 2020. Therefore, authors have to include few more latest references in the list so that the gap covered for last few years.
3. In the manuscript, U-Net model is used for the feature extraction. But in the given detail diagram of the U-Net is not properly drawn. In the figure, it is not clear that which layer is used for conv. operations and which layer is used for other. So redraw it and clearly mention that the above told issues.
4. Minor language editing required and Language proof read has to be done by the experts.
5. Authors have to include latest reference like https://doi.org/10.1007/s00530-020-00694-1, DOI: 10.3233/JIFS-189773
Minor language proof reading required.
Round 2
Reviewer 1 Report
revision well done